

# Social rank overrides environmental and community fluctuations in determining meat access by female chimpanzees in the Taï National Park, Côte d'Ivoire

Julia Riedel[1,2], Leo Polansky[1,3], Roman M. Wittig[1,2] and Christophe Boesch[1,2]

[1] Department of Primatology, Max Planck Institute for Evolutionary Anthropology, Leipzig, Germany
[2] Taï Chimpanzee Project, Centre Suisse de Recherches Scientifiques, Abidjan, Ivory Coast
[3] Bay-Delta Fish and Wildlife Office, US Fish and Wildlife Service, Sacramento, CA, United States of America

Corresponding author
Julia Riedel, riedel@eva.mpg.de

## ABSTRACT

Meat, long hypothesized as an important food source in human evolution, is still a substantial component of the modern human diet, with some humans relying entirely on meat during certain times of the year. Understanding the socio-ecological context leading to the successful acquisition and consumption of meat by chimpanzees (*Pan troglodytes*), our closest living relative, can provide insight into the emergence of this trait because humans and chimpanzees are unusual among primates in that they both (i) hunt mammalian prey, (ii) share meat with community members, and (iii) form long-term relationships and complex social hierarchies within their communities. However, females in both human hunter-gatherer societies as well as chimpanzee groups rarely hunt, instead typically accessing meat via males that share meat with group members. In general, female chimpanzee dominance rank affects feeding competition, but so far, the effect of female dominance rank on meat access found different results within and across studied chimpanzee groups. Here we contribute to the debate on how female rank influences meat access while controlling for several socio-ecological variables. Multivariate analyses of 773 separate meat-eating events collected over more than 25 years from two chimpanzee communities located in the Taï National Park, Côte d'Ivoire, were used to test the importance of female dominance rank for being present at, and for acquiring meat, during meat-eating events. We found that high-ranking females were more likely to be present during a meat-eating event and, in addition, were more likely to eat meat compared to the subordinates. These findings were robust to both large demographic changes (decrease of community size) and seasonal ecological changes (fruit abundance dynamics). In addition to social rank, we found that other female properties had a positive influence on presence to meat-eating events and access to meat given presence, including oestrus status, nursing of a small infant, and age. Similar to findings in other chimpanzee populations, our results suggest that females reliably acquire meat over their lifetime despite rarely being active hunters. The implication of this study supports the hypothesis that dominance rank is an important female chimpanzee property conferring benefits for the high-ranking females.

## INTRODUCTION

Hunting and meat eating are considered important behaviors shaping early hominid evolution and are proposed to be key innovations in the evolution of Australopithecines to *Homo erectus* (*Isaac, 1978*; *Washburn, 1978*; *Leakey, 1981*; *Hill, 1982*). In particular, meat consumption has for at least several decades been suggested as the food that powered brain expansion in human evolution (*Washburn & Lancaster, 1968*; *Milton, 1999*). Meat has become a substantial component in many modern human diets, recently averaging 42 kg of meat per capita per year (*Faostat, 2014*).

Placed within the context of human evolution, observations of hunting and meat eating in our closest living relatives, chimpanzees (*Pan troglodytes*), are critical in helping to reconstruct early human behaviors (*Wrangham, 1987*; *Boesch-Acherman & Boesch, 1994*; *Gilby et al., 2017*). Since early-published records of chimpanzee hunting and meat eating by *Goodall (1963)*, numerous other field studies have reported these as common chimpanzee behaviors (*Boesch & Boesch, 1989*; *Nishida, 1990*; *Uehara et al., 1992*; *Stanford, 1999*; *Mitani & Watts, 1999*; *Boesch & Boesch-Achermann, 2000*). So although chimpanzee diets are composed primarily of fruits (*Goodall, 1968*; *Sugiyama & Koman, 1992*; *Morgan & Sanz, 2006*), meat is considered an important food source year round. The nutritional value of meat is not easy to substitute because of its high quality calorific package of protein, fat and micronutrients that are difficult to find in plant foods (*Stanford, 1999*; *Milton, 2003*; *Tennie, Gilby & Mundry, 2009*).

The frequency of meat eating varies considerably across chimpanzees groups, sex and individuals. In Taï National Park in Côte d'Ivoire (hereafter referred to as Taï), chimpanzee meat eating varies with the seasons (wet/dry) and over the years, but is an important activity all year round, taking up to 9% of their activity budget, with Taï males and females eating an average of 186 g and 25 g of meat per day, respectively (*Boesch & Boesch-Achermann, 2000*). A stable isotope study of hair and bones confirmed sex differences in meat consumption, with higher levels of meat eating among Taï chimpanzee males compared to females (*Fahy et al., 2013*). For Gombe chimpanzees (Tanzania), where similar to Taï regular hunting occurs, an average of 22 g of meat per day is reported (*Wrangham, 1975*), with males eating an average of 55 g and females 7 g (*Boesch & Boesch-Achermann, 2000*). At Ngogo (Uganda), the estimates for daily meat intake are 41–55 g for males and 10 g for females. This big chimpanzee group hunts a lot with a total annual biomass of 850 kg meat (*Watts & Mitani, 2002*). At Fongoli (Senegal), females seem to hunt more compared to other study sites, accounting for 30% of all captures and males for 70% (*Pruetz et al., 2015*). Here, the female meat intake might be higher compared to the other study sites. The Fongoli chimpanzees sometimes hunt with tools; females hunt more with tools than males and prefer to hunt *Galagos* (*Pruetz et al., 2015*). There is a considerable variation in the frequency of hunting and the amount of meat intake across the studied chimpanzee groups, in all groups, males

hunt more than females and also eat more meat (*Goodall, 1986*; *Stanford, 1999*; *Boesch & Boesch-Achermann, 2000*; *Watts & Mitani, 2002*; *Gilby et al., 2017*).

Social rank can also play an important role in the access and consumption of meat (*O'Malley et al., 2016*, but see *Samuni et al., 2018b* that found no effect of rank on meat access). A study of Gombe Kasekela chimpanzees showed that females use their dominance rank position to maximize their access to meat, with high-ranking females consuming more meat than subordinates (*O'Malley et al., 2016*). Further, it has been shown that meat is an important food source for female chimpanzees during periods of pregnancy and nursing (*O'Malley et al., 2016*), a nutritionally and energetically costly body state for females (*Clutton-Brock & Harvey, 1978*; *Emery Thompson, Muller & Wrangham, 2012*). The Gombe study found an interaction between reproductive state and rank, and revealed that high-ranking females do not differ in meat consumption between different reproductive states, but low-ranking females do, with low-ranking females consuming more meat during pregnancy than during lactation and baseline (not pregnant/ not lactating females) (*O'Malley et al., 2016*). In the Gombe Kasekela group, high-ranking females have a higher hunting probability compared to lower ranking females, but this was not true at the Gombe Mitumba group and Kanyawara group in Kibale, Uganda (*Gilby et al., 2017*). In Sonso chimpanzees of the Budongo forest in Uganda, high-ranking individuals monopolize meat irrespective of their own hunting role, whereas in the neighbouring Waibira community no rank effect on meat access was found (*Hobaiter et al., 2017*). Dominance rank has been shown to affect sociality, ranging, and feeding competition in female chimpanzees at Gombe (*Pusey, Williams & Goodall, 1997*; *Pusey et al., 2005*; *Williams, Liu & Pusey, 2002*; *Williams et al., 2002*; *Murray, Eberly & Pusey, 2006*; *Murray, Mane & Pusey, 2007*) and Kibale (*Emery Thompson et al., 2007*; *Emery Thompson et al., 2010*; *Kahlenberg, Emery Thompson & Wrangham, 2008*).

As chimpanzees live in fission–fusion societies whereby individuals form smaller foraging parties when food abundance is low (*Boesch & Boesch-Achermann, 2000*; *Goodall, 1986*; *Mitani, Watts & Lwanga, 2002*), it is important to consider their gregariousness when studying how they access meat. The flexibility provided by fission–fusion grouping helps reduce within-group contest competition over food, and chimpanzee females tend to be less social then males, although differences between study sites have been documented (*Boesch & Boesch-Achermann, 2000*; *Fawcett, 2000*; *Wakefield, 2002*; *Williams, Liu & Pusey, 2002*; *Lehmann & Boesch, 2008*). Taï females are more gregarious compared to those from Kibale and Gombe (*Wrangham & Smuts, 1980*; *Goodall, 1986*; *Wrangham, Clark & Isabirye-Basuta, 1992*; *Pusey, Williams & Goodall, 1997*; *Williams, Liu & Pusey, 2002*; *Lehmann & Boesch, 2008*). Furthermore, it has been shown that high-ranking females are more gregarious with other females than are low-ranking females in Gombe (*Williams, Liu & Pusey, 2002*). At Taï, a high rank appears to allow females to be more gregarious in times of low fruit abundance, whereas during seasons of high fruit abundance, all females were highly gregarious, regardless of their rank (*Riedel, Franz & Boesch, 2011*). When fruits were scarce, low-ranking females decreased their gregariousness, whereas high-ranking females' social behaviour changed little, revealing the social benefits of high rank (*Riedel, Franz & Boesch, 2011*).

Taï males are primarily the hunters; they capture the prey in most cases, and subsequently share it with their community members (*Boesch & Boesch-Achermann, 2000*; *Samuni et al., 2018b*). Taï females participate in hunts much less than males, with their involvement in hunting being approximately 13%–15% (*Boesch & Boesch, 1989*). That chimpanzee females can also be successful hunters has been shown from Fongoli, where about 40% of all successful hunters were females and the two high-ranking adult females were among the top 10 hunters (*Pruetz et al., 2015*). Because hunting participation can largely determine meat access (*Boesch & Boesch-Achermann, 2000*; *Samuni et al., 2018b*), the consumption of meat is not evenly distributed between community members, resulting in some individuals frequently not receiving meat while others obtain it regularly (*Boesch & Boesch, 1989*; *Boesch, 1994*; *Watts & Mitani, 2002*; *Boesch & Boesch-Achermann, 2000*).

In the absence of hunting participation, "cheating", defined as consuming meat from a successful hunt that an individual did not participate in, can occur. By cheating, females are able to increase their caloric intake without suffering the energetic costs related to hunting. Females are more successful at cheating than males because male hunters tolerate female cheaters much more than male cheaters (*Boesch & Boesch-Achermann, 2000*).

In Taï, male hunters share meat unevenly with females, and males might share with cheating females because females copulated more with males who shared meat with them than with males who did not share meat with them, irrespective of characteristics such as male rank, female rank, or age (*Gomes & Boesch, 2009*). In contrast, three other chimpanzee study sites did not find support for the meat-for-sex hypothesis: at Ngogo, Gombe and Kanyawara the males did not gain mating advantages through meat sharing (*Mitani & Watts, 2001*; *Gilby, 2006*; *Gilby et al., 2010*).

Despite the low frequency in which females participate in hunting, some high-ranking females in the Taï North group can reach a very high status in the meat access order although they did not participate in the hunt, occasionally even surpassing low and middle ranking males (*Boesch & Boesch-Achermann, 2000*). In the Taï North group, high-ranking females monopolize and possess the food after a dyadic female/female food conflict and meat was the main reason for contests over monopolizable food among Taï females (*Wittig & Boesch, 2003*). Recent results investigating all food sharing events in Taï South and East groups found a three-way interaction between rank of the beggar and the possessor with sex on food access through sharing (*Samuni et al., 2018a*), indicating that rank neighbours were sharing more than dyads with a larger rank difference. When investigating access to meat only, however, another study on the same chimpanzee groups found an impact of hunt participation, age, prey size and fruit availability on meat consumption, but none by sex or dominance rank (*Samuni et al., 2018b*).

Here we utilize a large, long-term (greater than 25 years) dataset to detect the importance of different social factors on female success to access meat. The central two hypotheses tested here are that in comparison to lower ranking females, high-ranking females are more likely to (1) be present during a meat eating event, and to (2) have increased access to meat. The primary predictor variable of interest here is female dominance rank, while controlling for age, oestrous status and the nursing of a small infant, important factors in determining feeding competition more broadly (*Riedel, Franz & Boesch, 2011*; *O'Malley et*

*al., 2016*) and hence important control variables to include when testing for the importance of rank. We added socio-ecological control variables such as community size and fruit abundance to test for the robustness of rank effects besides seasonal ecological changes and large demographic changes (decrease of community size).

## MATERIALS & METHODS

### Study site and data collection

We analysed meat-eating data from two habituated chimpanzee communities in Taï, labelled North and South due to the relative geographic locations of their territories. Habituation of the North and South groups started in 1979 and 1989, respectively. Researchers and local field assistants have since continuously observed both communities. During the study period, the North group decreased from 76 to 19 individuals and the South group from 56 to 39 chimpanzees (Fig. 1), largely related to disease outbreaks and poaching.

Human observers carried out daily focal animal follows (*Altmann, 1974*) using standardised check-sheets and shifted to *ad libitum* data recording for hunting and meat eating events. Inter-observer reliability among Taï assistants is very good (*Deschner et al., 2004*; *Riedel, Franz & Boesch, 2011*). During the daily focal animal follows, the observers make continuous records of social interactions, party composition, and number and identity of females in oestrous encountered by the focal chimpanzee throughout the observation day. From these focal data, we were able to identify female dominance rank and the oestrous status. At hunting and meat eating events, the assistants changed the focus from the focal animal to the whole hunting and meat-eating situation and recorded as much information as possible about all visible individuals and interactions between them. At meat eating events, observers noted which chimpanzees were present, who held the prey, dyadic membership in meat sharing events, who was begging, who ate meat, who was aggressive and received aggression, as well as consumption time and prey details such as species and age-class. From the *ad libitum* data, we obtained information on whether a female was present within the meat-eating party, and whether the female got and ate meat. This includes all data whenever a female was eating meat, independently if she received the meat in whatever way, through sharing, begging or stealing, or when she had captured herself a prey. We were not able to include the amount of meat eaten by the female, because the data did not include this information.

The dataset covers a 27-year period for the North group (1987–2014) with 451 meat-eating events, and a 15-year period for the South group (1999–2014) with 376 meat-eating events. Due to logistical constraints, not all meat-eating events during this period were recorded. Our meat eating analyses focused on adult females, defined to be 13 years or more (*Boesch & Boesch-Achermann, 2000*), with 39 adult females from the North group and 33 females from the South group. Females in Taï give birth to their first infant when they are approximately 13 years old and at this age they are also fully grown and defined to be an adult (*Boesch & Boesch-Achermann, 2000*).

From the 827 observed meat-eating events, 773 had at least one adult female present and these were used for analysis. Analysis of data for which at least one female from each

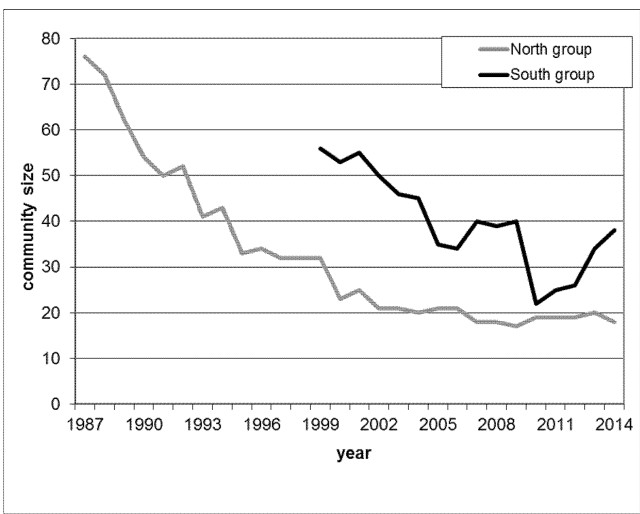

**Figure 1** Yearly maximal community size during the study period for North group (grey line) from 1987 until 2014 and for South group (black line) from 1999 until 2014.

rank category was present ($N = 464$) did not differ from the results using the full dataset; thus we present results based on the entire 773 events dataset. The meat eating events were almost entirely on separate days (94%) with a mean interval of 17 days (range of 1–254) for the South group and 25 days (range 1–289) for the North group.

Assistants trained in botanical monitoring censused fruit tree phenology every month using established routes in both chimpanzee territories. They noted the presence of ripe fruits of tree species whose fruits were chimpanzee foods (*Anderson et al., 2002*; *Anderson et al., 2005*; *Polansky & Boesch, 2013*).

All field protocols, data collection procedures, and data analyses were conducted in accordance with wildlife research protocols and ethical standards of the Max Planck Society in Germany, ''Ministère de l'Enseignement supérieur et de la Recherche scientifique'', ''Ministère des Eaux et Forêts'', and ''Office Ivoirien des Parcs et Réserves'' in Côte d'Ivoire.

## Data analysis, model predictors and motivations

We constructed two models for the response variables of (i) whether a female was present or absent in the meat eating party (hypothesis 1) and (ii) whether the female did or did not receive meat (hypothesis 2). Both models therefore have a bivariate response variable. We considered both 'individual female' properties and 'socio-ecological' properties as predictor variables, with interactions between some of these predictors, which we describe next.

### Dominance rank

According to *Wittig & Boesch (2003)*, we expected linear dominance hierarchies for the adult females in Taï. Using the software package MatMan (*De Vries, 1995*), we determined

annual linear dominance hierarchies following the direction of greeting behaviour: pant-grunts (PG), greeting-hoohs (GH) and greeting-pants (GP) (*Wittig & Boesch, 2003*). Only 6 out of 44 annual dominance hierarchies were significantly linear. The reasons why we rarely detected linear female hierarchies are a high percentage of unknown dyadic dominance relationships between females and years with just four adult females in the North group. Due to this, we implemented three rank categories (high, middle and low) following the method used in Gombe by *Pusey, Williams & Goodall (1997)*. We determined rank categories as follows. High-ranking females either did not give greetings to any females or gave occasional greetings to other high-ranking females and received greetings from middle- and low-ranking females. Middle-ranking females gave greetings to high- and some middle-ranking females, and received greetings from low- and some middle-ranking females. Low-ranking females rarely, if ever, received greetings from any adult females but often gave them to middle- and high-ranking females. When there was no greeting behavior observed between a female dyad in a certain year, we considered the ranks and interactions between these females in the year before and after. Furthermore, we consulted other rank data published about the Taï females (*Boesch & Boesch-Achermann, 2000*; *Wittig & Boesch, 2003*) and have always been able to assign females to one of the three categories. Female rank categories remained stable across years in both communities, with 58 of 80 females maintaining a single rank category over the study period. Twenty-two females moved to the adjacent category, mostly from low to middle (10 females) and from middle to high (10 females). Only two females (one in each community) dropped in rank from high to middle during the last years before death, as their physical condition deteriorated. A recent study from Taï also found that female dominance hierarchies of both the North and South group were largely stable over time and only few rank changes were described (*Mielke, Crockford & Wittig, 2019*).

### Nursing a small infant

We controlled whether the female was nursing a small infant ($\leq$ 2 years old) on the day of the meat-eating event because several studies have shown that chimpanzee mothers are less gregarious (*Goodall, 1986*; *Takahata, 1990*; *Sakura, 1994*; *Wrangham, 2000*; *Williams, Liu & Pusey, 2002*; *Otali & Gilchrist, 2006*; *Murray, Mane & Pusey, 2007*; but also see *Riedel, Franz & Boesch, 2011*). Mothers with small infants might avoid meat-eating events for the protection of their infants because of the large party sizes and competitive interactions to access meat at these events (*Boesch & Boesch-Achermann, 2000*). Alternatively, nursing females may disproportionately benefit from the nutritive value of meat, so it is also plausible to predict that they would try to join meat-eating events at higher frequencies. Males might prefer to share meat with mothers and their infants (potentially their own offspring) as a provisioning strategy. In Gombe chimpanzees, the reproductive state of females influenced meat consumption, with pregnant females consuming more meat than lactating and not pregnant/ not lactating females (*O'Malley et al., 2016*). Furthermore, it has been shown in Gombe and Kanyawara that females with a small infant do not avoid hunting; and females were equally likely to hunt red colobus whether or not they had an offspring under two years of age (*Gilby et al., 2017*).

### Oestrous status

We controlled for oestrous state as females with a maximal sexual swelling are more gregarious (*Boesch & Boesch-Achermann, 2000*) and adult males tend to share more meat with oestrous than with anoestrous females in Taï given their proportional representation in hunting parties (*Gomes & Boesch, 2009*). The same result has been found in Ngogo (*Mitani & Watts, 2001*: but see *Gilby, 2006*; *Gilby et al., 2010* for no effect in Gombe and Kanyawara). Assistants recorded the oestrous status, which coded sexual skin swellings after visual inspection following *Furuichi (1987)*. Three stages of tumescence were coded: (1) no swelling: minimal size and maximal degree of wrinkling; (2) partial swelling: relative increase/decrease in size and loss/appearance of wrinkles compared with stage 1 or 3; (3) maximum swelling: maximum size with no wrinkles and tight appearance. For the analysis we used whether the female had a maximal swelling at the meat-eating event.

### Age

We controlled for the age of the adult female although *Wittig & Boesch (2003)* found that female linear hierarchy in Taï was related to the outcome of the contest, while it was independent of age. Other studies in Gombe and Mahale found that females increased their rank as they aged (*Nishida, 1989*; *Pusey, Williams & Goodall, 1997*). Older Taï chimpanzees gained more access to meat (*Samuni et al., 2018b*).

### Number of females and males

We controlled for the number of adult females and males present at the meat-eating event because an increase in competitors also increases the within-group contest competition over food (*Wittig & Boesch, 2003*), although sub-group size previously had no effect on meat consumption in Taï East and South groups (*Samuni et al., 2018b*).

### Community ID

We controlled for potential differences across the two chimpanzee communities (*Luncz, Mundry & Boesch, 2012*) and included to which community the female belonged (either North or South group).

### Community size

Demography is a potential major driver of hunting behaviour whereby in large communities hunting frequency increases as well as the number of hunters acting together (*Mitani & Watts, 1999*; *Boesch & Boesch-Achermann, 2000*; *Watts & Mitani, 2002*). Both affect the amount of meat available within one community. Since over the study period, the community sizes of both communities decreased dramatically (Fig. 1), and for South group later increased again, we included into our analysis monthly community size to test for the potential demographic effect that could importantly affect the role of dominance on securing food and female meat access. Community size for each of the two communities was recorded at monthly intervals and the size recorded closest to the meat-eating event was used.

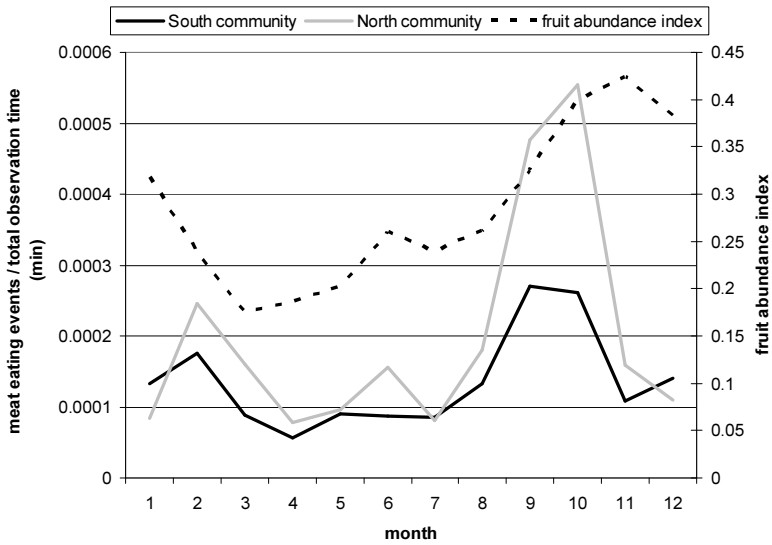

**Figure 2** **Monthly variation in the meat eating frequency controlled for the total observation time (min) for North and South group with a peak in September and October.** The second Y-Axis shows the monthly fruit abundance index (FAI) with an increase in FAI during the two meat eating peak months.

### Fruit abundance index

Chimpanzee hunting behavior and meat eating frequency in Taï is seasonal and peaks in September/October each year (*Boesch & Boesch-Achermann, 2000*) at the beginning of a major increase in the monthly fruit abundance index (FAI) (Fig. 2; see also *Polansky & Boesch, 2013*). In contrast, a recent study on the Taï chimpanzees found that meat accessibility increased with a decrease in fruit availability (*Samuni et al., 2018b*). Therefore, we considered a monthly fruit abundance index (FAI) as an ecological predictor variable for female presence at meat-eating events and meat access. To obtain a monthly FAI we used the monthly percentage of observed trees presenting ripe fruits eaten by the Taï chimpanzees following *Wrangham, Conklin-Brittain & Hunt (1998)*.

### Interactions between some predictors

Cultural differences between the North and South groups have been described (*Luncz, Mundry & Boesch, 2012*), so we included an interaction of female dominance rank and community ID into both models. Because intra-annual dynamics of FAI are substantial (*Anderson et al., 2005*; *Polansky & Boesch, 2013*), we included an interaction between FAI and female dominance rank into both models. We included the interaction of number of females with rank into both models. Female dominance rank might have less of an influence for presence at meat eating events if the number of adult females is low and therefore there is less competition. The same might be true for the access to meat, when there are many other female competitors at the meat-eating event, high rank might have a strong influence on the chances of accessing meat. Whereas when there is only a small number of other females present all of them independently of their rank might receive parts of the males share.

In addition, we included the interaction between female dominance rank and oestrous status into both models, because females in oestrous are very social and interesting for the males in Taï (*Deschner et al., 2004*), and independent of their rank they might be present and access meat. So not only high-ranking females in oestrus might get meat, but also low ranking females in oestrus.

## Statistical analyses

Generalized linear mixed models (GLMM) with a logit link function were built to analyze the significance of the factors described above. The GLMM framework accommodates the bivariate data (indicating either female presence or absence or whether or not she received meat, given presence at a meat eating event), while the mixed structure (both fixed and random effects) allows proper treatment of repeated measurements on individual females and meat eating events due to unobserved variables (e.g., total amount of meat available). Fixed effect predictor variables were adult female dominance rank, age, oestrous status, whether the individual was nursing an infant, community ID, FAI, the number of adult males present, and number of adult females present (community size was highly correlated with number of adult females in the presence model). Random effects grouping variables were the individual female ID and the meat-eating event ID. As discussed previously, for both models we included the interactions of dominance rank and FAI, dominance rank and community ID, dominance rank and number of adult females present, and dominance rank and oestrus status.

We fit models in the R version 3.5.2 environment (*R Core Team, 2018*) using the lme4 package (*Bates et al., 2015*) and followed the general guidelines described by *Bolker et al. (2009)*. The general model building process proceeded by first fitting a model with only random effects to ensure estimated standard deviations were not close to zero. For the presence model, this revealed numerical issues when random slopes were allowed for categorical predictor variables (oestrus, nursing an infant, and dominance rank), so these were removed prior to estimating a null model with only an intercept and supported random effects. For the meat access model, no random slopes were included to avoid numerical issues. Given the null model consisting only of an intercept, random effects, and an overall variance estimate, the full model including main effects and interactions along with random effects was compared to the null model using a likelihood ratio test (LRT). Interaction terms were removed one at a time from the full model and their statistical support quantified using LRTs. Because the interactions were not statistically supported, reduced models that excluded all interaction terms was re-fit to facilitate direct interpretation of the coefficient estimates, and the support for each of these main effects was evaluated using LRTs by removing each predictor variable one at a time.

Prior to fitting the full GLMMs, we standardized continuous predictor variables to have mean = 0 and standard deviation = 1 to increase model fitting stability. Furthermore, multicollinearity was checked by examining the generalized variance inflation factors (GVIF; *Zuur et al., 2009*) as implemented in the car package (*Fox & Weisberg, 2011*) in a full model without interactions or random effects. This indicated that community size and number of females were too correlated (GVIF = 21.094) to simultaneously include in the

presence model. For both models, residual correlations within individual were investigated in two ways using functions available in the stats package of R (*R Core Team, 2018*): (i) to make sure there was no autocorrelation using the acf function, and (ii) to see if there was a trend as a function of the number of days since the last time the individual was included in the analysis using the lm function. In each case, there was no evidence of problematic residual patterns. We remark that there was no strong evidence (GVIF values always less than 2) that female age and dominance rank were correlated.

## RESULTS

Seventy-two different adult females were observed, two of which never were present at any meat-eating event. Of the 773 events, at least one low, middle, and high-ranking adult female attended 530 (69%), 677 (88%), and 697 (90%) events. The number of events at which at least one female of a given rank received meat given that rank class was present was 329 (62%), 556 (82%), and 619 (89%) for low, middle and high rank class, respectively.

### Probability of females being present at meat eating events

The full model (including interactions) for the probability of being present at meat eating events was significantly better than a null model with only random effects and an overall intercept (test statistic 234.331, $df = 17$, $P$-value < 0.001). All proposed interactions were not significant (Table 1). The reduced model refit without including these interactions was also significantly better than the null model (test statistic 224.385, $df = 9$, $P$-value<0.001) but not worse than the full model (test statistic 9.946, $df = 8$, $P$-value = 0.269).

Parameter values and individual term significance of the reduced model are shown in Table 1. This indicates that female dominance rank, nursing a small infant, oestrus status, and the increased fruit abundance (FAI) have significant positive influence on the probability of females being present at a meat eating event. Increased number of males' present at the meat-eating event resulted in a decrease in the probability of females being present. Not significant was age, number of females, and community ID.

### Probability of females accessing meat given presence at a meat-eating event

The full model (including interactions) for the probability of accessing meat given presence at a meat eating event was significantly better than a null model with only random effects and an overall intercept (test statistic 166.142, $df = 18$, $P$-value < 0.001). All proposed interactions were not significant (Table 2). The reduced model refit without including these interactions was also significantly better than the null model (test statistic 158.007, $df = 10$, $P$-value <0.001), but not worse than the full model (test statistic 8.135, $df = 8$, $P$-value = 0.420).

Parameter values and individual term significance of the reduced model are shown in Table 2. This indicates that female dominance rank, age, nursing a small infant, and oestrus status have significant positive influence on the probability of females accessing meat, while increased community size and increased number of females statistically decreased the likelihood to obtain meat. The number of males, FAI, and community ID were not significant.

**Table 1  Summary of models for the probability of being present at meat eating events.** The term estimates columns show the estimate (Est), standard error (SE), and Z-value for model parameters, where the entries along the rows that do not include interactions (denoted by a colon) are based on the fitted reduced model with no interactions and the entries for rows with interactions are based on the fitted full model with all interactions and main effects. The term significance entries show results of likelihood ratio tests between either the full (when testing the importance of an interaction) or reduced model (when testing the importance of a predictor variable in isolation) and a model with the corresponding term removed.

| Terms | Term estimates | | | Term significance | | |
|---|---|---|---|---|---|---|
| | **Est** | **SE** | **Z-value** | **$\chi^2$** | **df** | **P-value** |
| Intercept | −1.080 | 0.222 | −4.871 | – | – | – |
| **Dominance rank (high-middle)** | **1.086** | **0.199** | **5.449** | **32.255** | **2** | **<0.001** |
| Dominance rank (middle-low) | 0.425 | 0.163 | 2.613 | – | – | – |
| Age | 0.194 | 0.112 | 1.736 | 2.735 | 1 | 0.098 |
| **Nursing a small infant (yes)** | **0.243** | **0.075** | **3.250** | **10.312** | **1** | **0.001** |
| **Oestrous status (yes)** | **1.234** | **0.102** | **12.097** | **150.842** | **1** | **<0.001** |
| **Fruit Abundance Index (FAI)** | **0.157** | **0.062** | **2.539** | **6.325** | **1** | **0.012** |
| **Number of males** | **−0.340** | **0.099** | **−3.444** | **11.275** | **1** | **0.001** |
| Number of females | −0.181 | 0.103 | −1.760 | 2.915 | 1 | 0.088 |
| Community ID (South) | −0.190 | 0.273 | −0.694 | 0.481 | 1 | 0.488 |
| Dominance rank (high-middle): No. of females | −0.240 | 0.156 | −1.533 | 4.342 | 2 | 0.114 |
| Dominance rank (middle-low): No. of females | −0.288 | 0.138 | −2.095 | – | – | – |
| Dominance rank (high-middle): FAI | 0.088 | 0.082 | 1.079 | 1.971 | 2 | 0.373 |
| Dominance rank (middle-low): FAI | 0.116 | 0.082 | 1.412 | – | – | – |
| Dominance rank (high-middle): Community ID (South) | 0.069 | 0.436 | 0.159 | 1.898 | 2 | 0.387 |
| Dominance rank (middle-low): Community ID (South) | 0.459 | 0.399 | 1.149 | – | – | – |
| Dominance rank (high-middle): Oestrous status (yes) | 0.199 | 0.260 | 0.764 | 0.998 | 2 | 0.607 |
| Dominance rank (middle-low): Oestrous status (yes) | −0.044 | 0.236 | −0.186 | – | – | – |

**Notes.**
$\chi^2$, test statistic;  df,  degrees of freedom.
Significant model parameters are marked bold.

## DISCUSSION

We found support for our two principal hypotheses, that high-ranking females were more likely to be present during a meat-eating event and, when present, they were more likely to eat meat compared to the subordinates. This research contributes to a growing body of literature on the topic, where high female rank has also been shown to provide priority of access to high quality foods in Kibale and Gombe chimpanzees (*Murray, Eberly & Pusey, 2006*; *Murray, Mane & Pusey, 2007*; *Kahlenberg, Emery Thompson & Wrangham, 2008*; *Williams et al., 2002*).

For chimpanzees at Taï, our analyses further indicated that this positive effect of female dominance rank on acquiring meat was stable for the studied chimpanzee communities over a long period (more than 25 years). During this time, both communities experienced large declines in size with associated demographic changes that include different numbers of adult females and adult males, individual identities, relationships and friendships lost from one day to the next and important changes in the communities' dominance hierarchies. In addition, the intra-annual fruit food fluctuations in Taï are quite strong and have been increasing in the past decade

**Table 2   Summary of models for the probability of meat access given presence at an event.** See the caption of Table 1 for details on entries.

| Terms | Term estimates | | | Term significance | | |
|---|---|---|---|---|---|---|
| | Est | SE | Z-value | $\chi^2$ | df | P-value |
| Intercept | −0.080 | 0.212 | −0.376 | – | – | – |
| **Dominance rank (high -middle)** | **1.090** | **0.208** | **5.232** | **26.206** | **2** | **<0.001** |
| Dominance rank (middle - low) | 0.707 | 0.171 | 4.138 | – | – | – |
| **Age** | **0.402** | **0.094** | **4.286** | **18.708** | **1** | **<0.001** |
| **Nursing a small infant (yes)** | **0.293** | **0.083** | **3.514** | **11.999** | **1** | **0.001** |
| **Oestrous status (yes)** | **0.459** | **0.138** | **3.335** | **10.9** | **1** | **0.001** |
| Fruit Abundance Index (FAI) | 0.091 | 0.070 | 1.300 | 1.655 | 1 | 0.198 |
| Number of males | −0.046 | 0.090 | −0.507 | 0.259 | 1 | 0.611 |
| **Number of females** | **−0.321** | **0.077** | **−4.184** | **17.57** | **1** | **<0.001** |
| **Community size** | **−0.193** | **0.095** | **−2.030** | **4.001** | **1** | **0.045** |
| Community ID (South) | −0.068 | 0.231 | −0.295 | −0.043 | 1 | 1 |
| Dominance rank (high-middle): No. of females | 0.080 | 0.114 | 0.698 | 1.804 | 2 | 0.406 |
| Dominance rank (middle-low): No. of females | 0.149 | 0.113 | 1.321 | – | – | – |
| Dominance rank (high-middle): FAI | 0.160 | 0.109 | 1.465 | 2.385 | 2 | 0.303 |
| Dominance rank (middle-low): FAI | 0.144 | 0.109 | 1.327 | – | – | – |
| Dominance rank (high-middle): Community ID (South) | −0.200 | 0.393 | −0.511 | 0.45 | 2 | 0.798 |
| Dominance rank (middle-low): Community ID (South) | 0.027 | 0.339 | 0.078 | – | – | – |
| Dominance rank (high-middle): Oestrous status (yes) | 0.585 | 0.353 | 1.658 | 2.87 | 2 | 0.238 |
| Dominance rank (middle-low): Oestrous status (yes) | 0.399 | 0.341 | 1.170 | – | – | – |

(*Anderson et al., 2005*; *Polansky & Boesch, 2013*). Despite these changes in demographic and environmental conditions, no interactions between female social rank and the different socio-ecological variables were detected; the rank related behaviors of these female chimpanzees are both stable and robust across the two communities. Further, the rank contribution to meat access remained constant across all these social changes.

In humans, sharing of food has been proposed to be the result of a collective action problem due to living in a risky foraging niche that produces a set of social norms of production and sharing (*Jaeggi & Gurven, 2013*). In other words, humans live in a niche where food sharing became a necessity. Considering that meat brings along a number of important micronutrients (*Milton, 2003*; *Tennie, Gilby & Mundry, 2009*), it seems that meat, acquired through the presence of sharing, is an important component of the diet in Taï chimpanzees.

Two recent studies in Taï, over a shorter period of time, focusing on dyadic interactions or detailed hunting characteristics, have shown that social rank independent of sex has only a limited effect in the food sharing behaviour of the Taï chimpanzees or their ability to access meat after a hunt (*Samuni et al., 2018a*; *Samuni et al., 2018b*). In contrast, focusing on the characteristics of the females, here we found a clear effect of rank on meat access by females, with dominant females accessing meat more often than subordinates. At least three reasons may account for this difference: (1) since we used data over 25 years the analysis had to fit the available long-term data, preventing us from analysing for example,

the effects of hunt participation or reciprocal relationships on meat access; (2) our research question was different and we did not include dyadic relationships; and (3) due to missing dyadic dominance relationships, we used three rank categories, which may provide a slightly different picture compared to linear dominance hierarchies.

Our finding that female dominance rank affects accessibility of valuable resources is consistent with findings that high rank confers advantages during contest competition over food (*Wittig & Boesch, 2003*). In Taï, females contest over food, being dominant over a competitor provided an advantage, as dominant conflict partners possessed the food significantly more frequently after conflicts than did subordinates independent from the initiator (*Wittig & Boesch, 2003*). Therefore, our research points to the importance of rank for acquiring meat, the causative mechanisms by which rank confers these benefits remain to be studied. Research on food sharing in Taï has shown that mutual grooming relationships predict best with whom to share food (*Samuni et al., 2018a*). Therefore, it could be that dominant females are more successful in building and maintaining social relationships with males. In contrast, studies in Gombe have shown that harassment best predicts the outcome of meat sharing (*Gilby, 2006*). Therefore, an alternative hypothesis would be that dominant females could be better in harassing males than subordinates, although harassment has not been shown to be a strong predictor of food sharing in Taï (*Samuni et al., 2018a*). Finally, in Taï dominant females invest more in sons (*Boesch, 1997*) and therefore sons might share more meat with their mothers, making kin relationships a main predictor for meat sharing. Other important variables to be investigated further could include begging, stealing, female hunting, dyadic associations, kinship and grooming (*Gilby, 2006*; *Gilby et al., 2010*; *Samuni et al., 2018a*; *Samuni et al., 2018b*).

In Gombe and Kibale, high ranking females occupied higher-quality areas while subordinates had to settle elsewhere (*Murray, Eberly & Pusey, 2006*; *Murray, Mane & Pusey, 2007*; *Kahlenberg, Emery Thompson & Wrangham, 2008*; *Williams et al., 2002*) resulting in higher reproductive success for dominant females (*Pusey, Williams & Goodall, 1997*; *Pusey et al., 2005*; *Emery Thompson et al., 2007*). A better-fed female can invest more energy in reproduction and thereby produce more offspring, or she can supply more food to her offspring. In Gombe, the five most successful females at getting large amounts of meat had more surviving offspring than did the five least successful females (*McGrew, 1992*).

The effect of female dominance rank on feeding competition appears across chimpanzee populations besides differences in demography and female sociality (*Wrangham, 2000*; *Fawcett, 2000*; *Boesch & Boesch-Achermann, 2000*; *Williams, Liu & Pusey, 2002*; *Williams et al., 2002*; *Lehmann & Boesch, 2008*). Taï females are more gregarious compared to those from Kibale and Gombe where females were relatively asocial (*Wrangham & Smuts, 1980*; *Goodall, 1986*; *Wrangham, Clark & Isabirye-Basuta, 1992*; *Pusey, Williams & Goodall, 1997*; *Williams, Liu & Pusey, 2002*). Females in Kibale and Gombe seem to disperse and to be less gregarious to reduce within-group contest competition. The higher gregariousness in Taï females may result from a combination of higher fruit abundance (*Boesch, 2009*) and higher predation pressure at Taï (*Boesch, 1991*; *Boesch, 2009*) compared to the other study populations. Unequal access to monopolizable food, such as meat, might be an explanation for the development of the linear hierarchy in Taï females (*Wittig & Boesch, 2003*), which

we were only able to find in some of the years we studied. Female dominance rank might help to reduce direct dyadic fighting by giving access to the dominant individual before even a conflict or fight have to evolve.

*Samuni et al. (2018b)* found a positive effect of age on meat access in Taï chimpanzees, which we confirmed. It remains unclear why older chimpanzees are more likely to access meat. It may be due to better begging and/or hunting skills. One hypothesis is that older females can have stronger friendships with chimpanzees in their community and can rely on long-term cooperative exchanges that gives them access to shared foods such as meat (*Samuni et al., 2018a*). In addition, older females are also preferred mating partners by male chimpanzees (*Muller, Emery Thompson & Wrangham, 2006*).

Another female property proposed to play a role for presence and meat access was the oestrous status of the female. We found that oestrous females in Taï were more likely to be present and to get meat than females with no oestrous. This agrees with the findings that oestrous females were more gregarious than anoestrous ones (*Boesch & Boesch-Achermann, 2000*), and that adult males share more meat with oestrous than with anoestrous females, when controlled for their proportional representation in hunting parties (*Gomes & Boesch, 2009*).

Our results show that a nursing female with a small infant in Taï did not avoid meat eating events where elevated levels of intra-group aggression can occur. The increased need for high value food such as meat to support nursing an infant is a likely factor motivating these females to acquire meat. This goes in line with the findings from Gombe and Kanyawara were females with a small infant did not avoid hunting (*Gilby et al., 2017*). Chimpanzee mothers in Taï remained as gregarious as non-mothers (*Riedel, Franz & Boesch, 2011*), in contrast to other study populations, where mothers are less gregarious than non-mothers (*Goodall, 1986*; *Murray, Mane & Pusey, 2007*; *Williams, Liu & Pusey, 2002*; *Takahata, 1990*; *Sakura, 1994*; *Wrangham, 2000*; *Otali & Gilchrist, 2006*). Our results about lactating females accessing meat better than non-lactating females, confirms the findings from Gombe, where pregnant females consumed more meat than lactating and not pregnant/not lactating females (*O'Malley et al., 2016*). An interaction between reproductive state and rank in Gombe females, revealed that high-ranking females do not differ in meat consumption between different reproductive states, but low-ranking females do, with low-ranking females consuming more meat during pregnancy than during lactation and not pregnant/ not lactating females (*O'Malley et al., 2016*).

For the ecological variable that we studied, we found that fruit abundance had no significant effect on female meat access, but plays a role on female presence at meat eating events, with more females being present during times of high fruit abundance. That those periods of high fruit abundance result in higher party sizes and sociality in Taï chimpanzees has been shown before (*Doran, 1997*; *Riedel, Franz & Boesch, 2011*).

In Taï, females hunt much less than males, involvement in being 13%–15% (*Boesch & Boesch, 1989*; *Samuni et al., 2018b*). Nevertheless, we can report that adult females continued hunting and meat eating behaviors also during years when there were no or just one adult male in the North group. The North group had no adult males for four years and only one adult male for another six years, but a minimum of four adult

females that continued to hunt and ate meat during this period. Although hunting frequencies seemed reduced, it is impressive that females engaged successfully in this behavior, further supporting the evidence that meat has a high nutritional value for chimpanzees. Impressively, low-ranking females and even nursing mothers joined these aggressive meat-eating events and were successful in accessing meat, which strengthens further the importance of meat in the female chimpanzee diet.

## CONCLUSION

The benefits of female dominance rank for accessing meat are positive and robust to fruit abundance variations and large demographic changes and hence some group level social changes such as in dominance hierarchies. Taken together this indicates that this female social property is persistent even when the competition for resources declines because of overall community size declines or fruit abundance increases. Furthermore, other female properties such as age, oestrus status and the nursing of a small infant positively influenced meat access.

## ACKNOWLEDGEMENTS

We thank the Ministère des Eaux et Forêts, Ministère de l'Enseignement supérieur et de la Recherche scientifique, and the Office Ivoirien des Parcs et Réserves for permitting this research, and the Centre Suisse de Recherche Scientifique at Abidjan in Côte d'Ivoire for their logistical support; and the Max Planck Society. We thank the Taï Chimpanzee Project (TCP) and especially the TCP field assistants Camille Bolé, Nicaise Oulaï, Honora Kpazahi, Grégoire Nohon and Nestor Gouyan Bah, who collected the long-term data presented here. We are grateful to Roger Mundry for statistical advice and Robert Power for insights into human meat eating literature. This research was conducted in accordance with the animal care regulations and national laws of Côte d'Ivoire and Germany. Two reviewers and Editor Dr Antje Engelhardt provided important comments on an earlier version of this manuscript that substantially improved it. The viewpoints expressed are those of the authors and do not necessarily reflect the opinions of the U.S. Department of the Interior or the U.S. Fish and Wildlife Service.

### Funding

This work was supported by the Max Planck Society, which has provided core funding for the Taï Chimpanzee Project since 1997. The funders had no role in study design, data collection and analysis, decision to publish, or preparation of the manuscript.

### Grant Disclosures

The following grant information was disclosed by the authors:
Max Planck Society.

## Competing Interests

The authors declare there are no competing interests.

## Author Contributions

- Julia Riedel and Leo Polansky analyzed the data, conceived and designed the experiments, performed the experiments, prepared figures and/or tables, authored or reviewed drafts of the paper, and approved the final draft.
- Roman M. Wittig and Christophe Boesch conceived and designed the experiments, authored or reviewed drafts of the paper, and approved the final draft.

## Animal Ethics

The following information was supplied relating to ethical approvals (i.e., approving body and any reference numbers):

Ministère des Eaux et Forêts, Ministère de l'Enseignement supérieur et de la Recherche scientifique, Office Ivoirien des Parcs et Réserves and the Max Planck Society approved this research.

## Data Availability

The raw data and codes are available in the Supplemental Files.

## Supplemental Information

Supplemental information for this article can be found online at http://dx.doi.org/10.7717/peerj.8283#supplemental-information.

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
