# Peer review of "Social rank overrides environmental and community fluctuations in determining meat access by female chimpanzees in the Taï National Park, Côte d’Ivoire"

_PeerJ, doi:10.7717/peerj.8283_

## Round 0.1 · original submission · Major Revisions

While both reviewers valued the study, one of the reviewers had major concerns about how you approached the topic theoretically. As it seems, you have not set the study into comprehensive perspective, but left out quite some relevant and contradicting evidence. This needs to be included to give the reader an objective overview over the topic and to better understand your findings.

Reviewer 1 ·

Basic reporting

Overall: strong. I have made some suggestions for providing a more detailed context for the study by referencing some key papers on the topic (see General comments).

Re the raw data: It might help to provide a ‘legend’ for all the variables within each file (or as a separate text document to accompany all files), briefly explaining what they are and how they were coded.

Experimental design

Rigorous and appropriate. Very interesting study! (see some specific comments further below)

Validity of the findings

Given the large dataset originating from two separate chimpanzee communities, this is a robust analysis which has important implications for our understanding of female chimpanzee feeding ecology and how that interacts with social dynamics in the context of meat sharing. The discussion is clear and avoids the pitfall of over-speculation very well.

Additional comments

General comments

Line-by-line:
L32/3: check for sentence structure/grammar and paraphrase for clarity (‘the effect on meat access found different results’).

L47: ‘chimpanzee understanding’ is a bit ambiguous; the understanding of chimpanzee behavioural ecology (or whatever) would be clearer. Having said that as a final sentence of an abstract this is not very informative; may be instead of saying ‘we discuss findings and offer implications’ you could just state what is the main implication you have identified.

L63-65: When giving average amount consumed per year might be helpful to specify these data come specifically from Tai in the text. Otherwise it sounds like this applies to chimpanzees everywhere. It would also be good to acknowledge there is considerable variation in frequency of hunting and amount of meat intake across study sites.

L70. Please replace ‘gender’ with ‘sex’ (here and throughout the MS). With animals we cannot easily discern their gender, we can only observe their biological sex. This is an important distinction.

L79, 88 etc.: Here and throughout the MS and in the list of references please correct ‘Thompson’ to ‘Emery Thompson’ (no hyphen) for the last name of the relevant author.

L69-72: Here or elsewhere in the MS where relevant (perhaps in the Discussion) it would be useful to provide a more detailed review and comparison with data on sex differences in hunting/consumption to encompass observations from other study sites. E.g., new data on higher than usual hunting by females are available for Fongoli: Pruetz et al. 2015 Roy Soc Open Science.

L99: I find the use of the term ‘cheating’ potentially misleading and unnecessarily ambiguous (and also perhaps inaccurately anthropomorphic). I see no need to define individuals getting access to meat that they did not participate in capturing as ‘cheating’. The use of this word can be read as implying that they somehow ‘trick’ others. That’s hardly the case. This is very much a stylistic preference/issue but I do think the text will be clearer and more precise without this ‘term’.

L106-109: This statement, while technically correct for the case of Tai chimpanzees, skirts around the larger issue of the assumptions and validity of the ‘meat-for-sex’ hypothesis. It should be made more explicitly clear that this only applies to long-term patterns observed at Tai (as opposed to direct short-term tests of this hypothesis at Tai or all the tests at other study sites that have failed to support the hypothesis: Gombe & Kanyawara). Along these lines, the absence of a reference to the paper, which examines this idea at great detail, is puzzling: I think you need to address this gap here or in the Discussion by directly contrasting your findings to those from Gombe & Kanyawara (Gilby et al. 2010 J Hum Evol).

L126: The hypothesis that higher-ranking females are more likely to be present during a meat eating event (H1) is interesting but I think needs a stronger/clearer rationale. What is the mechanism you envisage here? Given the fission-fusion system of chimpanzees, the implication of your hypothesis is that simply higher-ranking females are more gregarious than low-ranking females, is it not? I think this is worth elaborating on so that the context of this hypothesis is clearer to readers. Are there comparable evidence to review from other study sites to suggest rank-differences in gregariousness/space use that you can use to frame this H1 in context? You discuss gregariousness in the Discussion but I think it would be useful to introduce the concept of fission-fusion and sex differences in gregariousness in the introduction when formulating your H1 (you don’t mention the ‘fission-fusion’ social system of chimpanzees at all in your entire paper – I think this is relevant background information).

L182: ‘We constructed two models’ is sufficient (no need for ‘and analysed’).

L192: Clarify what you mean when you say the hierarchies ‘reached significance’ (i.e. they were significantly linear?).

L214: ‘on the day’ (not ‘at the day’)

L276: Why could the rank effects differ between North and South – provide a rationale for including this interaction term, please.

L280: A more appropriate test for this would be to examine the variance inflation factors for the model (VIF). Which you have done later in the text (L324-232) – so why mention Pearson here?

L328/9: Could you clarify how you checked that specifically (e.g., which package/function?).

L375: Principal, not ‘principle’.

Reviewer 2 ·

Basic reporting

This paper tests the hypothesis that dominance rank in female chimpanzees increases their access to meat, which is interesting because male chimpanzees primarily hunt and possess meat, so females would have to get meat through sharing and/or tolerated theft. The authors find support that high-ranking females are more likely to be present when meat-eating occurs and are more likely to obtain a share.
This is an interesting article but has two major issues that should be addressed. First, while these analyses include lots of control variables, they are not carefully thought out with respect to their ability to test the hypotheses. The authors find that dominance rank influences female presence at meat-eating events, but they do not test whether high-ranking females are also simply more likely to be observed (e.g., if they are more gregarious) even when meat eating is not occurring. It was also not clearly stated specifically why they were testing this hypothesis. With respect to meat access, they authors find that dominance rank influences getting meat. This is certainly interesting, but the authors fail to provide any context for why this occurs. This should start with a clearer definition of what the outcome variable actually is because it is variably described as “got meat or not”, “did or did not receive meat”, “eating [or accessing] meat”. Did this refer simply to eating meat at all (i.e., could be by the female’s own capture) or does it mean receiving it from someone else? Prior studies of meat sharing in chimpanzees suggest that begging intensity is an important determinant of who gets meat. Since the authors collected data on begging, they should determine whether the rank effect could be driven by begging. Similarly, if sharing was the main thing that was accounting for female meat-eating, then it would be useful to determine whether kinship could explain some of this (i.e., high ranking females receiving meat from their sons). Throughout the discussion, the authors never do discuss explanations for HOW high-ranking females get more meat. There is a brief reference to contest competition, but is there evidence that these females are getting their meat by taking it from other females?
The second issue is that while the authors do cite studies of other chimpanzee populations for general things, they pointedly ignored highly-relevant literature on hunting and meat-sharing from other sites. In particular, the extensive work by Ian Gilby has not been cited at all, and there is only one passing reference to the work by Mitani & Watts. These scholars have made some important critiques and have made different conclusions, so ignoring their work yields a highly biased review.

Experimental design

see above

Validity of the findings

see above

Additional comments

Lines 61-65: I don’t immediately have access to the textbook cited here, but these numbers seem very high. “Up to 9% of their activity budget” is vague – does this refer to hunting? Or meat eating? Over what time frame? Surely they are not eating meat for >1 hr per day. This is probably meant to say that in some months, the chimpanzees spend up to 9% of their feeding time feeding on meat. The statement about caloric amounts also doesn’t make sense as written, so it must be per episode and/or perhaps across all individuals of each sex. By the data reported here, there are about 20 meat-eating episodes per year, so to get to 186g/day, males would have to eat 3.4 kg per episode!
Line 65-68. Grammatically incorrect. Please revise.
Line 70. Please use sex and not gender. Gender refers specifically to identity.
Lines 103-105. As written, this sentence contradicts itself with respect to whether females or males receive more
Lines 113-115. This sentence appears out of place as written
Lines 191-192. Please clarify which measure did not reach significance (e.g., I assume you mean linearity).
It seems ok to address the uncertainty of dominance ranks by assigning categorical ranks. However, the method for assigning high/middle/low is vague, and if the data were not sufficient to assign a linear hierarchy, it is not clear how well these will represent the data. Given the assumption that ranks are stable from year to year, I recommend to combine 2-3 years in order to generate more reliability in assessments. In the present method, it is unclear how many females fell into each category and if this varied a lot from year to year.
Line 218: This sentence appears to give a prediction rather than a declarative statement so should be reworded to make this clear.
Line 235: did this include pregnant females with swellings?
Line 250: in the interests of avoiding an overparameterized model, I suggest deleting the community size variable. The rationale given in the first sentence is really only about how many individuals can participate in hunting, thus the number of males present should already take care of this. With that variable and community ID, it is not clear what the significance of community size would be. [I see now that this was excluded from the first model…I suggest simply eliminating it entirely]
Line 264: clarify “accessibility”, as this sentence appears to contradict another sentence in this paragraph: do they hunt more with lower or higher FAI?
Line 271: correct “colleges” to “colleagues”.
Line 351: shouldn’t this be number of males present (not in the community)?
Lines 409-412: but there is no evidence that these females receive meat because of contest competition is there?
Lines 426-428: this sentence needs more explanation
Lines 426-430: but you’ve shown in this paper that the hierarchy is often not linear.

---

## Round 0.2 · accepted · Accept

Thank you for such a thorough and responsive revision. Your work represents considerable effort and rigor and I am happy to accept it for publication in PeerJ. You will note that I was not the acting editor on the initial submission but I have taken over to expedite the process. I apologize for the long delay in getting you a decision on this revision. I have only a very few minor corrections (below) that can easily be addressed during the proofing stage.

On line 113, change chimpanzees to chimpanzee
On line 120, place commas around “in all groups.”
On line 161, change reducing to reduce.
Place commas after e.g. and i.e. throughout.
On lines 276 and277, you do not need the “or not.”
On line 281, change “such” to “so.”
The sentence on lines 322-324 is awkward.
Line 414 place comma after “In contrast,..” and around “when present” on line 561 and “for example” on line 600-601, after Tai on line 619.
On line 487, change “were” to “where.”

Reviewer 1 ·

Basic reporting

Both the introduction and the discussion have been revised to provide detailed context and interpretation. A very strong revision!

Experimental design

Good as before!

Validity of the findings

Yes. As before - this is a strong study and an important contribution.

Additional comments

Thank you for this thorough revision. I think you've addressed all the comments well and the paper is now a much more comprehensive piece of writing, making the most of this amazing dataset.